# De Novo Cancer Mutations Frequently Associate with Recurrent Chromosomal Abnormalities during Long-Term Human Pluripotent Stem Cell Culture

**DOI:** 10.3390/cells13161395

**Published:** 2024-08-21

**Authors:** Diana Al Delbany, Manjusha S. Ghosh, Nuša Krivec, Anfien Huyghebaert, Marius Regin, Mai Chi Duong, Yingnan Lei, Karen Sermon, Catharina Olsen, Claudia Spits

**Affiliations:** 1Research Group Genetics, Reproduction and Development, Vrije Universiteit Brussel, Laarbeeklaan 103, 1090 Jette, Belgium; diana.al.delbany@vub.be (D.A.D.); manjusha.ghosh@vub.be (M.S.G.); nusa.krivec@vub.be (N.K.); anfien.emma.huyghebaert@vub.be (A.H.); marius.regin@vub.be (M.R.); chi.mai.duong@vub.be (M.C.D.); yingnan.lei@vub.be (Y.L.); karen.sermon@vub.be (K.S.); catharina.olsen@uzbrussel.be (C.O.); 2Department of Biochemistry, Military Hospital 175, 786 Nguyen Kiem Street, Ho Chi Minh City 71409, Vietnam; 3Brussels Interuniversity Genomics High Throughput Core (BRIGHTcore), Vrije Universiteit Brussel (VUB)-Université Libre de Bruxelles (ULB), Laarbeeklaan 101, 1090 Brussels, Belgium; 4Interuniversity Institute of Bioinformatics in Brussels, Université Libre de Bruxelles (ULB)-Vrije Universiteit Brussel (VUB), La Plaine Campus Triomflaan, 1050 Brussels, Belgium

**Keywords:** human pluripotent stem cells, genetic instability, copy number variation, single-nucleotide variants, cancer-related genes

## Abstract

Human pluripotent stem cells (hPSCs) are pivotal in regenerative medicine, yet their in vitro expansion often leads to genetic abnormalities, raising concerns about their safety in clinical applications. This study analyzed ten human embryonic stem cell lines across multiple passages to elucidate the dynamics of chromosomal abnormalities and single-nucleotide variants (SNVs) in 380 cancer-related genes. Prolonged in vitro culture resulted in 80% of the lines acquiring gains of chromosome 20q or 1q, both known for conferring an in vitro growth advantage. 70% of lines also acquired other copy number variants (CNVs) outside the recurrent set. Additionally, we detected 122 SNVs in 88 genes, with all lines acquiring at least one de novo SNV during culture. Our findings showed higher loads of both CNVs and SNVs at later passages, which were due to the cumulative acquisition of mutations over a longer time in culture, and not to an increased rate of mutagenesis over time. Importantly, we observed that SNVs and rare CNVs followed the acquisition of chromosomal gains in 1q and 20q, while most of the low-passage and genetically balanced samples were devoid of cancer-associated mutations. This suggests that recurrent chromosomal abnormalities are potential drivers for the acquisition of other mutations.

## 1. Introduction

hPSCs represent an invaluable source for regenerative medicine due to their capacity to differentiate into any cell type of the human body. Today, stem cell-based therapies are constantly advancing and have emerged as promising therapeutic avenues for patients suffering from degenerative diseases and cancers, with over 100 clinical trials currently ongoing [1]. At low passages, most hPSC lines maintain a normal diploid karyotype. However, during in vitro expansion, hPSCs frequently acquire genetic aberrations, a risk exacerbated by their infinite self-renewal capacity and pluripotency.

Draper et al. (2004) were the first to identify recurrent gains of chromosomes 12 and 17q in human embryonic stem cell (hESC) lines [2]. Numerous subsequent studies have confirmed this tendency, identifying the full or segmental gain of chromosomes 1, 12, 17, and 20 as the most recurrent genetic aberrations in both hESCs and human induced pluripotent stem cells (hiPSCs) [3,4]. Advancements in genomic screening, especially high-resolution microarray and sequencing techniques, have since enhanced CNV detection, enabling the identification of smaller CNVs and point mutations [5,6].

Despite extensive research, the mechanisms driving the high incidence of chromosomal abnormalities in hPSC cultures remains to be fully understood. However, it is generally accepted that the genetic abnormalities that are found in hPSC cultures originate from a single mutated cell that took over the culture thanks to a growth advantage provided by the mutation. hPSCs have several features that may contribute to their acquisition of genetic abnormalities. It is hypothesized that these abnormalities arise, at least in part, from the rapid progression of hPSCs through the cell cycle, which leads to DNA replication stress during proliferation [7,8]. They also have a shorter G1 phase and higher replication stress during the S phase than differentiated cells, resulting in increased DNA damage, double-strand breaks, and errors in repair mechanisms [9,10]. This stress, combined with impaired cell-cycle checkpoints and the decoupling of spindle assembly checkpoint mechanisms from apoptosis, can lead to an increased prevalence of chromosomal abnormalities [11,12]. Furthermore, hPSCs have more open chromatin, which may affect the efficiency of chromosome segregation, and have weaker centromeres, which may impair kinetochore recruitment [13]. Lastly, environmental stressors, like hypoxia and medium acidification [14,15,16], exacerbate DNA damage and contribute to chromosomal abnormalities. Overall, while the global mutation rate in hPSC division is relatively low [17], owing to these cells’ robust DNA damage response [18,19,20], all hPSC cultures exhibit some degree of low-grade mosaicism [16,21,22].

Chromosomal alterations may confer a growth advantage to the mutated cells [23]. This advantage is speculated to arise from the ability of the altered cells to avoid bottlenecks, such as increased cell death after plating and failure to re-enter the cell cycle, that normally restrict the expansion of genetically normal cells [24]. Interestingly, global variations in the prevalence of different CNVs across hPSC lines have been documented over time, which have been linked to changes in culture conditions that promote the dominance of different variants [25]. This highlights how selective pressures can favor the proliferation of certain mutants. Recent research has increasingly focused on understanding the mechanisms underlying these CNVs, their selective advantages, and their effects on differentiation. For instance, hPSCs with a gain of 20q11.21 take over the culture thanks to their higher resistance to apoptosis due to the additional copies of BCL2L1 [3,26,27,28]. In turn, this mutation impairs correct neuroectodermal differentiation, both in directed and spontaneous differentiation protocols [29,30]. An interesting study also revealed that hPSCs with multiple recurrent CNVs, have a selective advantage that is associated with mutant cells proliferating faster than wild-type cells and mechanically compressing the wild-type cells [31]. Worryingly, these genetic aberrations are reminiscent of those seen in cancers, raising concerns about the potential for oncogenic transformation in transplanted hPSC-derived cell products [6]. Nevertheless, it remains unclear whether the acquired genetic or epigenetic changes in hPSCs during their culture will affect the safety and the efficacy of hPSC derivatives produced for therapeutic applications [32]. Research has linked aneuploidy to the formation of more aggressive and metastatic teratomas, a type of tumor that can arise from pluripotent stem cells [33,34,35]. Therefore, precise testing for stem cell genetic stability is crucial to ensure that hPSCs are devoid of mutations that could render them or their differentiated progeny malignant when transplanted into patients [36].

Point mutations in cancer-related genes were first discovered in hiPSCs [37], but the origin and the biological effects of these mutations were unknown. Later, several research groups conducted genomic analyses on hPSC lines and investigated the incidence of point mutations in cancer-related genes [5,38,39,40]. Merkle et al., identified recurrent dominant negative mutations in TP53 in hESC lines, after analyzing the exome sequences of 140 lines, including lines that will be potentially used for clinical application [40]. The same group carried out whole-genome sequencing (WGS) of 143 hESC lines and revealed that a significant proportion of hESC lines had large deleterious structural variants, as well as finer-scale structural variants and SNVs associated with cancer and other diseases [5]. A subsequent study compared WES data of early-passaged cells to RNA-seq data from later passages of H1 and H9 (also known as WA01 and WA09), two of the most commonly used hPSC lines, and observed that cancer-related mutations are mostly acquired during prolonged culturing, predominantly in TP53 but also less frequently in other genes, including EGFR, CDK12, and PATZ1, genes that are deregulated in human tumors and have significant pathological potential [39]. A recent study from the same group analyzed over 2200 transcriptomes from 146 independent hPSC lines and found that 22% of the samples had cancer-related mutations, with 64% having TP53 mutations, that provided a selective advantage, disrupted target gene expression, and affected cellular differentiation [38].

While these studies provided valuable insights into the mutational landscape of hPSCs, they could not reliably determine the timing of the appearance of these variants or their association with other mutational events such as the acquisition of CNVs. In this work, we close this gap in the knowledge by studying multiple passages of ten hESC lines using simultaneous targeted gene sequencing with a panel of 380 cancer-associated genes and CNV analysis via shallow whole-genome sequencing to investigate the timing and association of these genetic variants. This approach provides an insight into the temporal evolution of genetic changes within each hESC line.

## 2. Materials and Methods

### 2.1. hESCs Lines and Cell Culture

All hESC lines were derived and characterized as reported previously [41,42,43]; details on the characterization of the lines can also be found at the open science framework https://osf.io/esmz8/ (accessed on 19 August 2024) and are registered in the EU hPSC registry https://hpscreg.eu/ (accessed on 19 August 2024). The cells were cryopreserved in freezing medium composed of 90% knock-out serum (Thermo Fisher Scientific, Ghent, Belgium) and 10% dimethyl sulfoxide (Sigma-Aldrich, Schnelldorf, Germany). In the past, our hESC lines were cultured on 0.1% gelatin-coated (Sigma-Aldrich, Schnelldorf, Germany) culture dishes containing mitomycin C-inactivated CF1 mouse embryonic fibroblast (MEF) feeders, in hES medium culture medium supplemented with 20% KO-serum replacement [43]. Cells were passaged through the manual dissection of undifferentiated cell colonies. In this study, hESCs were cultured on tissue culture dishes coated with 10 µg/mL Biolaminin 521 (Biolamina^®^, Sundbyberg, Sweden and maintained in NutriStem hESC XF medium (NS medium; Sartorius, Gottingen, Germany) with 100 U/mL penicillin/streptomycin (P/S) (Thermo Fisher Scientific, Ghent, Belgium). The cells were passaged as single cells using TrypLE Express (Thermo Fisher Scientific, Ghent, Belgium) and split in a ratio of 1:10 to 1:100 as needed at 70–90% confluence. The cells were fed daily with NutriStem hESC XF medium in a 37 °C incubator with 5% CO_2_. All cultures were tested monthly for the presence of mycoplasma. For this study, hESCs that had been cryopreserved from MEF cultures were thawed on Biolaminin-521 and NS medium and expanded for a few passages to obtain sufficient cells to extract DNA for the analysis. The identity of all samples in this study was authenticated by fingerprinting on the same DNA sample used for sequencing. Appendix A indicates which lines were kept on MEF prior to their cryopreservation and thawing for this study.

### 2.2. Fingerprinting

DNA fingerprinting was performed with a Devyser Complete v2 kit (Devyser, Hägersten, Sweden). Briefly, multiplex PCR was carried out, which interrogated 32 STR markers on chromosomes 13, 18, 21, X, and Y. Separation of the different amplicons was carried out on a Genetic Analyzer 3500 (ABI), and Genemapper v6 (Thermo Fischer Scientific, Ghent, Belgium) was used for subsequent data interpretation.

### 2.3. Whole-Genome Shallow Sequencing

The genetic content of the hESCs was assessed through shallow whole-genome sequencing (7 million reads of 50 bases, which corresponds to 0.1× coverage of the genome), by BRIGHTcore of UZ Brussels, Belgium, as previously described [44]. A total of 5 µL of purified DNA was processed using a KAPA HyperPlus kit (Roche Sequencing, San Francisco, CA, USA) according to the manufacturer’s recommendations, with five modifications: (1) enzymatic fragmentation for 45 min at 37 °C to obtain DNA insert sizes of, on average, 200 bp; (2) the usage of 15 µM of our in-house-designed UDI/UMI adapters (Integrated DNA Technologies, Coralville, IA, USA); (3) 0.8× post-ligation AMPure bead cleanup; (4) a total of 6 PCR cycles to obtain a sufficient library; and (5) 1× post-PCR AMPure bead cleanup. The final library was quantified and qualified with a Qubit 2.0 using a Promega Quantifluor ONE kit (Promega, WI, USA) and with an AATI Fragment Analyzer (Agilent Technologies Inc., Santa Clara, CA, USA) using a DNF-474 High-Sensitivity NGS Fragment Analysis Kit (Agilent Technologies Inc., CA, USA). The final library was diluted to 1.5 nM prior to denaturation for analysis on a NovaSeq S1 100 cycles run (Illumina Inc., San Diego, CA, USA), generating, on average, 7 million 2 × 50 bp reads. Following demultiplexing with bcl2fastq (v2.19.1.403), all reads were mapped to the human genome (UCSC b37) using BWA aln v.0.7.10. The aligned reads were sorted based on genomic coordinates using samtools sort (v0.1.19), and duplicates were removed with the Picard markduplicates tool (v.1.97). Following the removal of blacklisted regions (in-house table), the coverage was calculated in bins of 50 kb with the bed tool coverageBed (v2-2.25.0). Following GC correction and Z score, fold change, and log2ratio calculation using in-house-developed R scripts, the data were visualized in an in-house-developed tool called BRIGHTCNV. Part of the data visualization in BRIGHTCNV involved making use of JBrowse v1.0.1.

### 2.4. Gene Panels for Cancer-Associated Genes

A cancer-associated gene panel was internally designed (Appendix A), with a target average coverage of 1500× (https://www.brightcore.be/gene-panels/solid-tumors-and-haematological-tumors-(stht)---v3 (accessed on 19 August 2024). Libraries were constructed on 150 ng of input DNA with a KAPA HyperPlus kit (Roche Sequencing, CA, USA) according to manufacturer’s recommendations, with three modifications: (1) the use of enzymatic fragmentation for 20 min at 37 °C to obtain DNA insert sizes of, on average, 200 bp; (2) the usage of 15 µM of our in-house-designed UDI/UMI adapters; and (3) the application of a total of 8 PCR cycles to obtain a sufficient library for target enrichment. Target enrichment was performed according to version 5.0 of the manufacturer’s instructions with a homebrew (STHT v3) KAPA HyperChoice probemix (Roche Sequencing, CA, USA). Pre-capture pooling was limited to max. 8 samples for a total of 1.2 µg of pooled library. In contrast to the instructions, an xGen Universal Blockers TS Mix (Integrated DNA Technologies, Coralville, IA, USA) replaced the sequence-specific blocking oligos, and the final PCR was limited to 11 PCR cycles. The final libraries were qualified on the AATI Fragment Analyzer (Agilent Technologies Inc., Santa Clara, CA, USA) using the DNF-474 High-Sensitivity NGS Fragment Analysis Kit (Agilent Technologies Inc., CA, USA) and quantified on the Qubit 2.0 with the Qubit dsDNA HS Assay Kit (Life Technologies, CA, USA). Per sample, a minimum of 14.5 million 2 × 100 bp reads were generated on the Illumina NovaSeq 6000 system (Illumina Inc., San Diego, CA, USA), with the NovaSeq 6000 S4 Reagent Kit (200 cycles) kit. For this, 1 nM libraries were denatured according to the manufacturer’s instructions.

The raw basecall files were converted to .fastq files with the Illuminas bcl-convert algorithm. the Reads were aligned to the human reference genome (hg19) with bwa (version 0.7.10-r789). Duplicate reads were marked with picard (version 1.97). Further post-processing of the aligned reads was carried out with the Genome Analysis Toolkit (GATK) (version 3.3). This post-processing consisted of realignment around insertions/deletions (indels) and base quality score recalibration. Quality control on the post-processed aligned reads was performed with samtools flagstat (version 0.1-19) and picard HsMetrics (version 1.97). These tools were used to investigate the total number of reads, the percentage of duplicate reads, the mean coverage on target and the percentage of on-target, near-target, and off-target bases). Variants were called using GATK Mutect2 (version 4) in tumor-only mode and annotated using annovar (version 2018Apr16). Variants with a population frequency higher than 1% in gnomad, 1000 g, and esp6500 were filtered out. Furthermore, non-hotspot variants with an allele frequency below 3% were filtered out. These exclusion criteria were adopted based on the ComPerMed guidelines, which recommend a 0.1% population frequency threshold. However, we chose a 1% cutoff because at least one clinically relevant variable exceeded this frequency. The 3% allele frequency cutoff was selected because our panel was validated up to this level. Nonetheless, hotspot mutations can sometimes be detected at lower frequencies, so we included and further investigated these cases when they occured.

The called variants in the VCF files were visualized using the Basespace Variant Interpreter online tool https://variantinterpreter.informatics.illumina.com (accessed on 19 August 2024). Recurrent variants, defined as variants occurring in >0.01% of samples sequenced over time at the BrightCORE facility using the same gene panel, were filtered out. A variant read count of ≥25 was used as a cutoff to keep a variant in the analysis. After filtering, the variants were manually inspected on IGV (https://igv.org (accessed on 19 August 2024)). Any suspected genomic regions such as GC repeats, indels at microsatellite regions, etc., were removed from the analysis. The gene CDC27 was excluded from the analysis due to a high prevalence of pseudogenes [45].

## 3. Results and Discussion

In this work, we studied a total of 33 samples collected across 10 hESC lines: VUBe001 (passages 66, 117, and 285), VUBe002 (passages 6, 36, and 353), VUBe003 (passages 16, 61, and 98), VUBe004 (passages 16, 50, and 111), VUBe005 (passages 39, 75, and 89), VUBe007 (passages 26, 40, 88, 198, and 208), VUBe014 (passages 20, 50, and 88), VUBe019 (passages 26, 60, and 79), VUBe024 (passages 25, 43, and 75), and VUBe026 (passages 11, 35, 39, and 54). Figure 1A shows an overview of all the gains and losses identified, Figure 2 shows the genetic variants identified per cell line and passage, and a complete list of CNV breakpoints is provided in Appendix A. Of the 10 cell lines, only VUBe005 showed balanced genetic content at all the tested passages (P39, P75, and P89). One line (VUBe026) already carried CNVs at the earliest passage tested (P11), and the other nine lines acquired different chromosomal abnormalities during extended culture. Overall, gains were more common than losses (39 gains vs. 8 losses). We found that 80% of our lines (8/10 lines, 13/33 samples) acquired a gain of chromo-some 20q and 60% (6/10 lines, 8/33 samples) acquired 1q, both well-known highly re-current chromosomal abnormalities that confer a growth advantage to hPSCs in vitro [27,28,46,47]. All gains on 20q had a common proximal breakpoint and varying distal breakpoints, with sizes ranging from 1.075 Mb to 46.925 Mb, and included the driver gene BCL2L1, which encodes BCL-xl, an anti-apoptotic factor [48]. This gene is considered the driver of the mutation and is associated with a reduced capacity for ectodermal differentiation [27,28,29,30]. The proximal breakpoints for the gains of 1q were specific to each line and were telomeric, with their sizes ranging from 0.725 Mb to 103.7 Mb and spanning the driver gene MDM4, which diminishes cellular sensitivity to DNA damage by inhibiting p53-mediated apoptosis [25,46]. These findings fully align with previous reports on these recurrent abnormalities [3,5,48].

Two lines also carried losses of 18q, a recurrent but less common chromosomal abnormality in hPSCs [3,4,47,49]. The loss of chromosome 18q impairs neuroectodermal commitment, with the downregulation of SALL3, a gene located in the commonly lost region of 18q, being responsible for this compromised neuroectodermal differentiation [50]. Further, we found an array of other abnormalities, including duplications of 1p13.2, 1q21.3, 3q26.22q27.3, 7p22.3pter, 9p24.3p13.3, 15q26.1q26.2, and Xp11.3p11.23 and losses of 2q37.1, 6p21.33, 16p12.2p12.1, and 20p13, none of them being typically observed aneuploidies in hPSCs. These genetic changes do not appear to contain any genes with functions that would make them obvious driver genes for an in vitro selective advantage (listed in Appendix A). In all but one case, they appeared together with the recurrent genetic changes, suggesting that they may be passenger events.

The analysis of the datapoints of all lines combined shows that the later the sampling, the higher the load of acquired CNVs (*p* < 0.001, Poisson Loglinear Regression, Figure 1B). This association is not retained when considering the absolute passage number (*p* = 0.07, Figure 1C), but if the two latest passages are removed as outliers (passages 285 and 353), the association becomes statistically significant (*p* = 0.032). Conversely, later passages did not have a higher risk of acquiring a de novo CNV as such (*p* = 0.102, Binary Logistic Regression). This suggests that the higher loads of CNVs seen at later passages are due to the cumulative acquisition of mutations over a longer time in culture and not to an increased rate of mutagenesis over time.

Regarding the sequencing results, we identified 122 single-nucleotide variants (SNVs) in 88 of the 380 cancer-related genes examined (listed in Appendix A) across the different passages of the 10 hESC lines. These findings are detailed in Appendix A and summarized in Figure 2. While all 122 SNVs were different, 25 of the 88 genes were found to carry two or more SNVs. Since we sequenced samples of multiple passages of the same line, we could determine which of the SNVs were there from the onset of the establishment of the cell line (*N* = 96, assumed to be germline SNVs) and which appeared de novo during cell culture (*N* = 28). The allelic frequency of 95.8% of the detected germline SNVs (92 of 96) was around 0.5, as expected for heterozygous alleles; two of the germline SNVs were homozygous; and three SNVs had allelic frequencies of 0.25, 0.7, and 0.63 (Figure 1D). De novo SNVs had a peak in allelic frequency at around 0.5, but also frequently appeared with lower allelic frequencies, and once as homozygous (Figure 1E). All our hESC lines carried germline SNVs, ranging from 6 to 17 SNVs per line, with a similar distribution of the type of functional impact across lines (Figure 1F). From the 96 germline SNVs, 52 were missense SNVs, 29 of which were predicted to be deleterious mutations by the SIFT prediction tool (https://sift.bii.a-star.edu.sg/ (accessed on 19 August 2024). We also found 28 synonymous mutations, 1 stop-gain mutation, 5 SNVs at a splice site and 1 in the 5′ untranslated region, 2 in-frame deletions, and 7 mutations in introns (Appendix A and overview in Figure 2).

All hESC lines acquired at least one de novo SNV, with a maximum of five SNVs in the later passages (Figure 1G and Figure 2). Only two out of the ten lines showed de novo SNVs at the earliest passage tested (VUBe003 and VUBe004). Overall, the SNVs were predominantly found in samples that had been extensively passaged (mid and late passages), indicating that hESCs are more likely to acquire de novo variants during prolonged in vitro culture (*p* = 0.002, Poisson Loglinear Regression Figure 1H). The absolute passage number was not significantly associated with the number of de novo SNVs, even after removal of the outlier passage numbers (*p* = 0.101, Poisson Loglinear Regression, Figure 1I). Similarly, the rate of de novo mutagenesis did not increase with time in culture (*p* = 0.771 for passage number, *p* = 0.152 for sample rank, Binary Logistic Regression), suggesting that the SNV mutation rate stayed constant with time in culture. Overall, of the 40 de novo SNVs, 16 were missense mutations, of which 13 were predicted to be deleterious mutations, 7 synonymous changes, 12 stop-gain mutations, 1 a splice site region variant, 1 an in-frame deletion, and 3 SNVs in introns. None of the de novo SNVs we found to be progressively increased in allelic frequency across the passages we tested. While some SNVs appeared at low frequency in the earlier passages, their frequency over extended passaging did not change, suggesting that the cell fraction carrying the variant did not increase. Other variants decreased in frequency, or they disappeared entirely, in subsequent passages of the same line (Figure 2).

We categorized the mutations based on their potential deleterious effect, including stop gains and missense variants, and assessed their clinical relevance by checking if they had been previously reported in the Catalog of Somatic Mutations in Cancer (COSMIC: https://cancer.sanger.ac.uk/cosmic (accessed on 19 August 2024), irrespective of their tier. Remarkably, most of the identified germline and de novo variants were potentially deleterious, with no statistically significant differences between the two groups (57.3% (55/96) vs. 67.85% (19/28), respectively (two-tailed Fisher’s exact test, *p* = 0.3842, Figure 1J)). Likewise, 39.28% (11/28) of the de novo mutations had been reported in COSMIC, compared to 25% (24/96) of the germline mutations, which was also not statistically significant (two-tailed Fisher’s exact test, *p* = 0.1568, Figure 1J). We next investigated the similarities or differences between germline and de novo variants in terms of affected genes. Twelve genes carried both deleterious and non-deleterious germline SNVs, whereas in the de novo SNVs, the genes with deleterious mutations were unique and distinct from those with non-deleterious variants. Additionally, the germline and de novo SNVs had four genes in common (Figure 1K). Taken together, this shows that the germline and de novo SNVs differed in terms of potential functional impact. We then classified all the SNVs based on the cancer types with which the gene mutations are typically associated. The results show that there is not an especially enriched mutational profile associated with specific types of cancer in germline or de novo SNVs (Figure 1L). Lastly, we categorized the variants according to the gene’s function. Notably, de novo SNVs predominantly affected genes involved in transcriptional regulation and chromatin remodeling and were statistically significantly more often deleterious than their germline counterparts (*p* = 0.0154, Fisher’s exact test, Figure 1M).

Mutations in cancer-related genes have been previously identified in hPSCs [5,38,39,40]. These studies used various approaches to screen hPSC lines, focusing on Tier 1 COSMIC-reported variants. In a seminal report, Merkle et al. highlighted the recurrent acquisition of TP53 mutations in hPSCs [40]. Similarly, Avior et al. identified TP53 mutations as the most frequent in the H1 and H9 lines, along with mutations in EGFR, PATZ1, and CDK12 [39]. In a more recent large-scale follow-up study, Merkle et al. found 382 Tier 1 cancer-associated mutations across 143 lines via whole-genome sequencing, though they could not determine the mutation origins, since only a single sample per hPSC line was tested [5]. Lezmi et al. conducted an in-depth analysis of mRNA sequencing data and reported that 25% of the 146 hPSC lines they studied carry cancer-associated mutations [38]. In 70% of cases, they found that mutations appeared de novo in culture or during differentiation. Sporadic mutations in hPSCs can have significant phenotypic effects, limiting their utility in clinical applications. Amir et al. showed that hESCs with TP53 mutations gained a selective advantage under stressful culture conditions and retained a higher percentage of cells expressing the pluripotency marker OCT4 after differentiation, resulting in increased cell proliferation and survival rates [51]. Further, Lezmi et al. showed that TP53-mutated hPSCs had decreased neural differentiation capacities [38].

This study of multiple passages of the same line allowed us to establish with certainty if a variant appeared de novo, and when in time in culture this occurred, a data point that was missing in previously published work. We found that eight of the de novo and fourteen of the germline SNVs were Tier 1 COSMIC mutations. In total, 65% of our lines carried a Tier 1 COSMIC variant, 70% of them having acquired the variant in culture. When looking at which genes bear the mutations and the overlap with previous work, we also found that TP53 was most recurrent, but KMT2C was also recurrent (appearing twice in our study and identified by [5,38]). In our study, both TP53 variants were COSMIC-reported variants; one decreases in frequency with time in culture and the second is homozygous, suggesting a loss of heterozygosity. Other genes in common with previous reports are CREBBP, FAT1, PMS2, BRCA2, and APC, the last four appearing as germline variants in our study. BCOR mutations that have been reported in hiPSC [52] were not found in our study or previous work [5,38,39].

In the last step of the analysis, we integrated the CNV and SNV data, to test for associations between the two. In previously published work that tested for chromosomal abnormalities [5,39], identifying a link between CNV and SNV proved challenging because of either the relatively lower resolution of e-karyotyping [39] or the lack of multiple passages of the same line [5]. In our study, it is important to bear in mind that we did not always have a perfectly maintained single lineage within the lines, because some of the later passages of our lines were from historically frozen vials. Also, although some lines were maintained in culture continuously for years, they clearly drifted into genetically different sublines.

One initial observation from combining the CNV and SNV data is that the three germline variants, which had unexpected frequencies for being heterozygous or homozygous (as marked by a triangle in Figure 2), were actually located within a duplicated chromosomal region. VUBe026 carried at the earliest passage tested (Passage 11) had a duplication of 9p24.3p13.3. This region contains CDKN2B that carries an SNV with an allelic frequency of 0.7, as well as a deleterious and COSMIC-reported DOCK8 mutation with an allelic frequency of 0.25. The duplication of 9p24.3p13.3 explains the increased frequency of the CDKN2B SNV, as the line was likely heterozygous at the start, and the region with the variant was duplicated. The frequency of the DOCK8 mutation suggests that it may be a de novo mutation that occurred shortly after or concurrently with the gain of 9p24.3p13.3, and in linkage with the CDKN2B variant, explaining its frequency as a single copy in a triplicated locus. VUBe014 carried a PMS2 variant with an allelic frequency of 0.63 by passage 88 and simultaneously acquired an abnormal karyotype with a gain of the 7p arm, spanning the PMS2 gene.

A second and significant finding is that the majority of de novo SNV appears to follow the acquisition of chromosomal abnormalities, especially those known to confer a growth advantage to the cells. In total, 4 of the 13 karyotypically normal samples carried a de novo SNV, in contrast to 17 of the 20 karyotypically abnormal passages (30.7% vs. 85%, respectively; *p* = 0.0028, Fisher’s exact test). In all but three of the latter 17 instances, the abnormal karyotypes included gains of 20q11.21 and 1q, making it challenging to determine whether any of these SNVs individually confer a growth advantage to the cells or if they are merely passengers or complementary to the chromosomal abnormality. In VUBe024 and VUBe026 SNVs associated with CNVs, other than the gain of 20q or 1q, VUBe024 carried a small gain of 1q spanning genes with no obvious beneficial effect if duplicated, as well as a low-frequency deleterious variant in ASLX1. Both were lost in later passages, suggesting that they did not confer a growth advantage to the cells. VUBe026 is a more notable exception, where a gain of a deleterious variant in EP300 occurred between passages 11 and 35 and persisted in different sublines that acquired an array of different chromosomal abnormalities. This suggests that this specific variant may confer an in vitro advantage to the cells. By comparison, Avior et al. identified only one trisomy 17 concurrently with one of the TP53 mutations, but their method focused on detecting trisomies of 1, 12, 17, and 20, and would not have detected any of the gains of 20q found in our study, and only one of the gains of chromosome 1 [39].

In the large study by Merkle et al., half of the 14 SNVs identified as having the highest oncogenic potential were associated with aneuploidies [5]. In our study, five SNVs did not occur concurrently with CNVs, two were COSMIC-reported mutations (c.15461G > A in KMT2D (VUBe002) and c.2926G > A in KMT2C (VUBe003)), two were potentially deleterious (c.5934_5935dup in FAT3 (VUBe002) and c.238C > A in CYSLTR2 (VUBe005)), and the last one was a non-deleterious variant (c.4071G > C in MET (VUBe004)). The variant observed in CYSLTR2 was the sole abnormality acquired by the VUBe005 line at the latest passage tested, whereas the other variants disappeared in the later passage of the lines. It is possible that the mutations in the KMT2D and FAT3 genes persisted in an alternative subline of VUBe002, as they were detected at passage 36, with the next and final passage tested being passage 353, corresponding to a prolonged period in culture. Whether variants in these genes can have a role in promoting an in vitro growth advantage in hPSCs remains to be elucidated.

## 4. Conclusions

Our comprehensive analysis of the hESC lines across multiple passages provides insights into the dynamics of genomic alterations during in vitro culture. We observed the frequent emergence of both de novo CNVs and SNVs in cancer-related genes throughout culture, with mutation rates remaining stable over time, indicating that the higher mutational burden in later passages was cumulative due to prolonged culture rather than increased mutagenesis. Notably, the de novo SNVs often affected genes involved in transcriptional regulation and chromatin remodeling, with a higher proportion being deleterious and reported in COSMIC compared to germline SNVs. The integration of the CNV and SNV data revealed that CNVs that are not typically recurrent in hPSC cultures frequently emerge in association with known CNVs that confer a growth advantage to hPSCs, such as the gain of 1q and 20q11.21, and that many de novo SNVs appeared after the acquisition of these recurrent chromosomal abnormalities. These associations suggest two potential scenarios: either rare CNVs and the majority of the SNVs were passengers during the culture takeover of the recurrent CNVs or they potentially interacted and enhanced these CNVs. While functional experiments are necessary to fully understand their impact in regenerative medicine, it is rather reassuring that most of the low-passage and genetically balanced samples were devoid of de novo Tier 1 COSMIC mutations, ensuring their safety for use in research and therapeutic applications.

## Figures and Tables

**Figure 1 cells-13-01395-f001:**
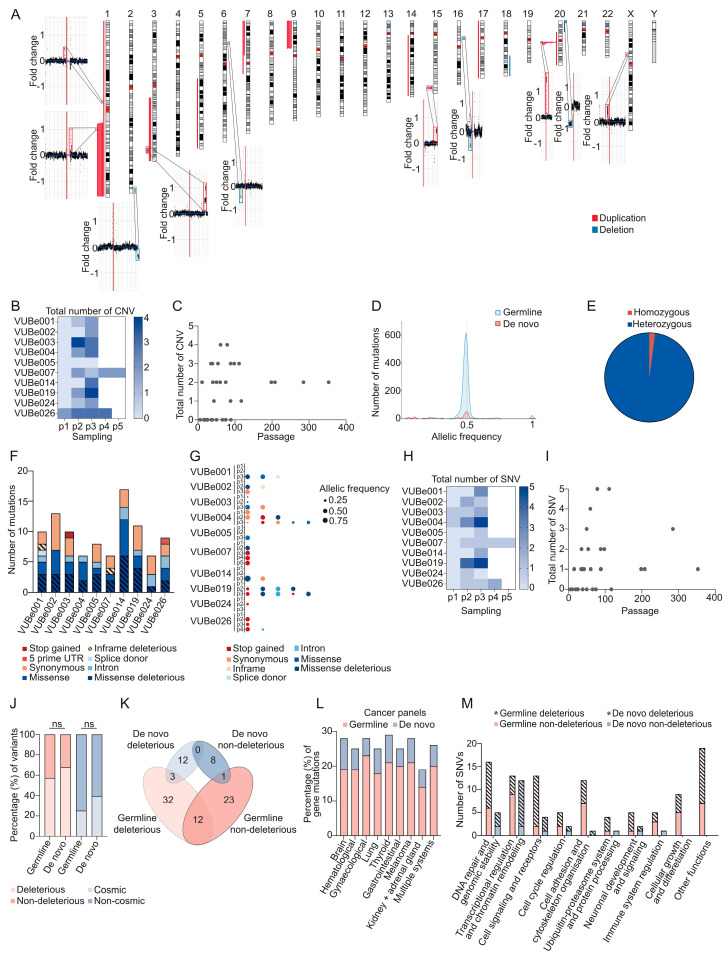
The CNVs and point mutations identified in the hESC samples from 10 different lines at multiple passages. (**A**) An ideogram showing the location of all gains (red) and losses (blue) found in the 10 hESC lines, as well as the shallow sequencing results of samples in which we found unique CNVs, such as gains on chromosomes 1, 3, 7, 9, 15, and X, and losses on chromosomes 2, 6, 16, and 20, that are not recurrent in hPSCs. (**B**) A heatmap of the total CNVs across different passages for each cell line. Each column (P1, P2, P3, P4, P5) represents the sequential sampling points (first, second, third, fourth, and fifth passages tested, respectively). (**C**) The total number of CNVs vs. the culture period in vitro across all the hESCs lines. Each dot represents a single cell line at a specific passage number. (**D**) Histogram representing the distribution of the allelic frequencies of all mutations grouped by mutation origin (germline, and de novo). (**E**) The distribution of the detected germline mutations by zygosity. The pie chart illustrates the proportion of homozygous (red) vs. heterozygous (blue) mutations identified in the samples. (**F**) The bars represent the distribution of mutation categories for germline SNPs and the number of events detected in each hESC line. (**G**) An overview of the de novo SNPs, the types of mutations detected in each passage of each hESC line, and the read frequency of each variant. Large bubble: high allelic frequency (0.75); average bubble: intermediate allelic frequency (0.50); small bubble: low allelic frequency (0.25); no bubble, no variant detected. (**H**) The heatmap shows the incidence and the number of de novo mutations found in hESC lines depending on their culture period in vitro. (**I**) The total number of SNVs vs. culture period in vitro across all the hESCs lines. Each dot represents a single cell line at a specific passage number. (**J**) The proportion of deleterious and non-deleterious variants (%) found among both the germline and de novo variants and whether they are reported in the COSMIC database. (**K**) A diagram showing the number of genes with germline (pink) or de novo (blue) mutations, and their pathogenic effect. (**L**) The distribution of gene mutations (%) across different cancer panels, categorized into germline (pink) and de novo mutations (blue). The *y*-axis represents the percentage of gene mutations, while the *x*-axis lists the various cancer types. (**M**) The bar graph illustrates the number of SNVs across various functional categories, differentiating between germline and de novo mutations, as well as their deleterious and non-deleterious effects.

**Figure 2 cells-13-01395-f002:**
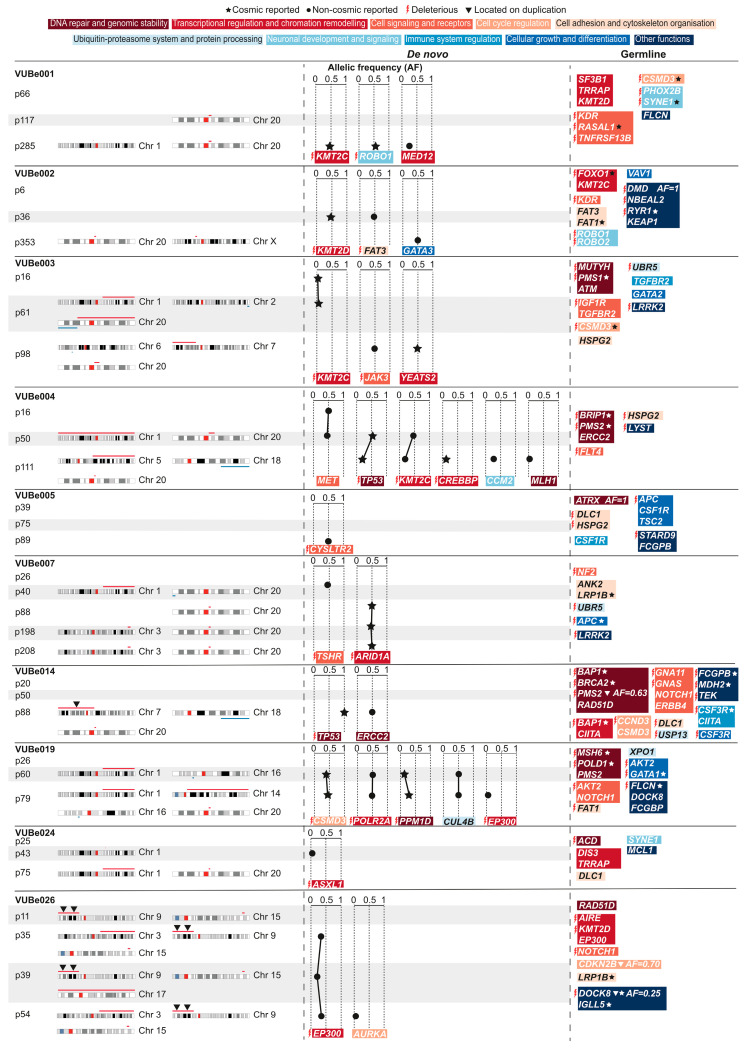
The distribution of chromosomal abnormalities and point mutations in cancer-related genes, with a focus on their functional impact in hESC lines. Chromosomal abnormalities acquired at a specific passage are shown for each hESC line, with the locations of all gains (red) and losses (blue) (**left section**). The point mutations found are shown in the middle (de novo) and right (germline) sections of the figure. Each mutation is annotated with the affected gene, the nature of the mutation, and its functional category, as shown by the function color key at the top of the figure. The allelic frequency (AF, scale 0 to 1) of de novo variations is indicated for each gene. Mutations reported in the COSMIC database are marked with ★, and mutations not reported in the database are marked with •. Mutations that are expected to be harmful (deleterious) are marked with 

. Mutations found in duplicated regions of the genome are marked with ▼.

## Data Availability

All VUB stem cell lines in this study are available upon request and after signing a material transfer agreement. Raw sequencing data of human samples are considered personal data by the General Data Protection Regulation of the European Union (Regulation (EU) 2016/679). Informed consent was obtained from all subjects involved in the study. The data can be obtained from the corresponding author upon reasonable request and after signing a data use agreement. The source data for all figures are available as a supplementary table. The data supporting all figures in this paper can be found at the Open Science Framework repository https://osf.io/wtmah (accessed on 19 August 2024).

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
