# Peer review of "De Novo Cancer Mutations Frequently Associate with Recurrent Chromosomal Abnormalities during Long-Term Human Pluripotent Stem Cell Culture"

_cells, 2024, doi:10.3390/cells13161395_

Round 1

Reviewer 1 Report

Comments and Suggestions for Authors

The manuscript by Delbany et al. report data on shallow whole-genome sequencing and targeted sequencing of a panel of 380 cancer-associated genes consisting of 33 samples collected from 10 hESC lines up to 353 passages. This is a significant work that complements a vast body of work on genetic abnormalities in pluripotent stem cells. Although the work is important, some weaknesses can be easily corrected.

How does the author define germline mutations in this work? Mutations present from the first sequencing? How to ensure that these are not de novo mutations that appeared before the first sequencing, during the derivation or the first passages?

It should be indicated whether the panel is a commercial kit (then indicate which one) or an internal design.

Report the average sequencing coverage of the WG shallow sequencing and cancer-associated gene panel.

The sentence “We studied a total of 33 samples collected across 10 hESC lines: VUBe001 (passages 66, 117, 285), VUBe002 (passages 6, 36, 353), VUBe003 (passages 16, 61, 98), VUBe004 (passages 16, 50, 111), VUBe005 (passages 39, 75, 89), VUBe007 (passages 26, 40, 88, 198, 208), VUBe014 (passages 20, 50, 88), VUBe019 (passages 26, 60, 79), VUBe024 (passages 25, 43, 75), and VUBe026 (passages 11, 35, 39, 54).” should be placed at the beginning of the results section.

The typos in Figures 1 and 2 are too small and need to be enlarged.

Fig 1G: allele frequency symbol is missing from legend

Line 346 is truncated

Author Response

The manuscript by Delbany et al. report data on shallow whole-genome sequencing and targeted sequencing of a panel of 380 cancer-associated genes consisting of 33 samples collected from 10 hESC lines up to 353 passages. This is a significant work that complements a vast body of work on genetic abnormalities in pluripotent stem cells. Although the work is important, some weaknesses can be easily corrected. 

 Comment 1: How does the author define germline mutations in this work? Mutations present from the first sequencing ? How to ensure that these are not de novo mutations that appeared before the first sequencing, during the derivation or the first passages? 

Response 1:  We defined germline mutations as genetic alterations that are present in the sample’s genome from the start, typically inherited from a parent and present in all cells. However, the concern raised is valid: mutations detected in the first sequencing could potentially include *de novo* mutations that occurred early in development or during the derivation or initial passages of the sample. As detailed in the text, we differentiated between SNVs present from the start of cell line establishment and those that emerged de novo during cell culture, and considered germline SNVs based on their allelic frequency. To confirm that these mutations are genuinely germline and not *de novo*, additional measures would be required, such as comparing the mutations with those from a reference sample of the same individual (e.g., blood or normal tissue). However, this approach is not feasible here, as the cell lines were derived from embryos donated for research. 

 Comment 2: It should be indicated whether the panel is a commercial kit (then indicate which one) or an internal design.  

Response 2: The cancer-associated gene panel is internally designed, with a target average coverage of 1500x (https://www.brightcore.be/gene-panels/solid-tumors-and-haematological-tumors-(stht)---v3). (Added to the material and methods) 

Comment 3: Report the average sequencing coverage of the WG shallow sequencing and cancer-associated gene panel. 

Response 3: For the shallow sequencing we aim at 7 million reads of 50 bases, which corresponds to a 0.1x coverage of the genome. (Added to the material and methods) 

Comment 4: The sentence “We studied a total of 33 samples collected across 10 hESC lines: VUBe001 (passages 66, 117, 285), VUBe002 (passages 6, 36, 353), VUBe003 (passages 16, 61, 98), VUBe004 (passages 16, 50, 111), VUBe005 (passages 39, 75, 89), VUBe007 (passages 26, 40, 88, 198, 208), VUBe014 (passages 20, 50, 88), VUBe019 (passages 26, 60, 79), VUBe024 (passages 25, 43, 75), and VUBe026 (passages 11, 35, 39, 54).” should be placed at the beginning of the results section. 

Response 4: Thank you for the suggestion. I agree that moving the sentence to the beginning of the results section would provide better context for the findings. I included this sentence at the start of the results section to improve the flow and clarity of the data presented. 

Comment 5: The typos in Figures 1 and 2 are too small and need to be enlarged. 

Response 5: Thank you for pointing this out. I enlarged the text in Figures 1 and 2 to ensure that it is clear and easily readable. 

 Comment 6: Fig 1G: allele frequency symbol is missing from legend  

Response 6: I added the following to the legend: Large bubble: high allelic frequency (0.75); average bubble: intermediate allelic frequency (0.50); small bubble: low allelic frequency (0.25); no bubble, no variant detected. 

 Comment 7: Line 346 is truncated  

Response 7: Thank you for bringing this to my attention. I corrected the truncated sentence on line 346.  

Reviewer 2 Report

Comments and Suggestions for Authors

In this study, researchers subjected human pluripotent stem cells to prolonged in vitro cell culturing. Whole genome sequencing was then used to identify mutations. The study has little novelty, as the results are exactly as expected. They found most cell lines acquired gains of chromosome arms known to confer in vitro growth advantages, and that random mutations accumulated over time.

Additional comments

Line 192 – 207: This paper is very descriptive. Therefore, it is easy to get bogged down in too much detail describing random mutations. It would be more helpful to the reader if the information in this section was present in the supplementary tables, and the main text were to provide a summary that points to the supplementary tables for the details.

Line 280. Sentence is clunky and should be rephrased.

Line 320. The resolution on Figures 1 and 2 appears low. It makes it difficult to read the small panels.

Author Response

In this study, researchers subjected human pluripotent stem cells to prolonged in vitro cell culturing. Whole genome sequencing was then used to identify mutations. The study has little novelty, as the results are exactly as expected. They found most cell lines acquired gains of chromosome arms known to confer in vitro growth advantages, and that random mutations accumulated over time. 

Comment 1: Line 192 – 207: This paper is very descriptive. Therefore, it is easy to get bogged down in too much detail describing random mutations. It would be more helpful to the reader if the information in this section was present in the supplementary tables, and the main text were to provide a summary that points to the supplementary tables for the details. 

Response 1: Thank you for the suggestion. The current text does provide an overview of the findings and directs readers to supplementary table 2 and Figure 2 for detailed information. I reviewed this section to ensure that the summary is as concise as possible.  

Comment 2: Line 280. Sentence is clunky and should be rephrased.  

Response 2: Thank you for the feedback. I rephrased the sentence on line 280 to improve its clarity and flow. 

Comment 3: The resolution on Figures 1 and 2 appears low. It makes it difficult to read the small panels.  

Response 3: Thank you for pointing this out. I enlarged the text in Figures 1 and 2 to ensure that it is clear and easily readable. 

Reviewer 3 Report

Comments and Suggestions for Authors

Delbany and colleagues present a screen of mutations evaluated in human pluripotent stem cells subjected to extensive in vitro culture and cryopreservation. The manuscript needs a revision in its format, as suggested below:

A) Lines 32 to 75: A super long paragraph makes reading confusing and tiring. Break it down into smaller paragraphs.

B) Line 153: Authors should include a brief justification for adopting such exclusion criteria.

C) Results: Apparently the "Results" section was written as "Results and Discussion". If this possibility is contemplated in the journal's rules, then rename this section to the most appropriate nomenclature.

D) Lines 164 to 191, 245 to 271, and 280 to 319: A series of very long paragraphs makes reading difficult. Adjust.

E) Figure 1: Increase the size of the figure to allow for better reading.

F) Line 346: Incomplete sentence.

G) Lines 348 to 374: Text with different formatting than the rest of the manuscript.

H) Line 358: Renomear para "Conclusões".

Author Response

Al Delbany and colleagues present a screen of mutations evaluated in human pluripotent stem cells subjected to extensive in vitro culture and cryopreservation. The manuscript needs a revision in its format, as suggested below: 

Comment 1: Lines 32 to 75: A super long paragraph makes reading confusing and tiring. Break it down into smaller paragraphs.  

Response 1: Thank you for your feedback. I broke down the long paragraph on lines 32 to 75 into smaller, more manageable paragraphs to make the text easier to follow. 

Comment 2:  Line 153: Authors should include a brief justification for adopting such exclusion criteria.

Response 2:  The exclusion criteria were adopted based on the ComPerMed guidelines, which recommend a 0.1% population frequency threshold. However, we chose a 1% cutoff because at least one clinically relevant variable exceeded this frequency. The 3% allele frequency cutoff was selected because our panel was validated up to this level. Nonetheless, hotspot mutations can sometimes be detected at lower frequencies, so we include and further investigate these cases when they occur. (Added to the material and methods) 

Comment 3: Results: Apparently the "Results" section was written as "Results and Discussion". If this possibility is contemplated in the journal's rules, then rename this section to the most appropriate nomenclature.  

Response 3: I revised the section title to align with the journal MDPI *Cells* guidelines. The journal allows for a combined "Results and Discussion" section, so we will retain the current title.  

Comment 4: Lines 164 to 191, 245 to 271, and 280 to 319: A series of very long paragraphs makes reading difficult. Adjust.  

Response 4: Thank you for your feedback. I broke down the long paragraphs into smaller, more manageable paragraphs to make the text easier to follow. 

Comment 5: Figure 1: Increase the size of the figure to allow for better reading.  

Response 5: Thank you for pointing this out. I enlarged the text in Figures 1 and 2 to ensure that it is clear and easily readable. 

Comment 6: Line 346: Incomplete sentence.  

Response 6: Thank you for bringing this to my attention. I corrected the incomplete sentence on line 346. 

Comment 7: Lines 348 to 374: Text with different formatting than the rest of the manuscript.  

Response 7: Thank you for pointing this out. I fixed the formatting.  

Comment 8: Line 358: Rename to "Conclusions"  

Response 8: I renamed the section to "Conclusions" as suggested.